

# Marine snow morphology drives sinking and attenuation in the ocean interior

**List of Authors:**

Yawouvi Dodji Soviadan[1,9,10*] ORCID 0000-0003-0622-5772

Miriam Beck[1] 0000-0001-8179-3820

Joelle Habib[1] 0000-0001-9604-7927

Alberto Baudena[1] ORCID 0000-0002-8890-4324

Laetitia Drago[1, 2] ORCID 0000-0002-0054-1734

Alexandre Accardo[1] ORCID 0009-0008-9426-2250

Remi Laxenaire[3,8] ORCID 0000-0001-5157-1821

Sabrina Speich[4] ORCID 0000-0002-5452-8287

Peter Brandt[5,6] ORCID 0000-0002-9235-955X

Rainer Kiko[1,5,6] ORCID 0000-0002-7851-9107

Stemmann Lars[1,7*] ORCID: 0000-0001-8935-4531

Sorbonne Université, CNRS, Laboratoire d'Océanographie de Villefranche, UMR 7093 LOV, Villefranche-sur-Mer, France
Sorbonne Université, UMR 7159 CNRS-IRD-MNHN, LOCEAN-IPSL, Paris, France
Laboratoire de l'Atmosphère et des Cyclones, LACy, UMR 8105, CNRS, Université de La Réunion, Météo-France, Saint-Denis de La Réunion, France
Laboratoire de Météorologie Dynamique - IPSL, ENS - PSL, Paris, France
GEOMAR Helmholtz Centre for Ocean Research Kiel, Kiel, Germany
Faculty of Mathematics and Natural Sciences, Kiel University, Kiel, Germany
Institut Universitaire de France, France
Center for Ocean-Atmospheric Prediction Studies, Florida State University, Tallahassee, FL, USA
Université de Lomé (TOGO)
MARBEC, IRD, IFREMER, CNRS, Université de Montpellier, 87 Avenue Jean Monnet, 34200 Sète, France

* Corresponding authors: Yawouvi Dodji SOVIADAN (syawouvi@yahoo.fr or yawouvi_dodji.soviadan@ird.fr ) and  Lars STEMMANN ( lars.stemmann@imev-mer.fr )





**Abstract**
Simultaneous measurements of marine snow (particles larger than 600 µm) morphologies, estimates of
their *in situ* sinking speeds and midwater attenuation in export plumes were performed for the first time using a
BGC-Argo float equipped with optical and imaging sensors. The float was deployed and recovered after one year
drifting in the sluggish flow regime of the Angola basin. Six consecutive chlorophyll-a and particulate matter
accumulation events were recorded at the surface, each followed by an export plume of sinking aggregates.
Objects larger than 600 µm were classified using machine learning recognition and clustered into four
morphological categories of marine aggregates. Plankton images were validated by an expert in a few broad
categories. Results show that different types of aggregates were produced and exported from the different blooms.
The different morphological categories of marine snow had different sinking speeds and attenuation for similar
size indicating the effect of morphology on sinking speed. However, the typical size-to-sinking relationship for
two of the categories and over the larger observed size range (100 µm-few mm) was also observed, indicating the
importance of size for sinking. Surprisingly, calculated *in situ* sinking speeds were constantly in the lower range
of known values usually assessed *ex situ*, suggesting a methodological effect which is discussed. Moving away
from purely size-based velocity relationships and incorporating these additional morphological aggregates
properties will help to improve mechanistic understanding of particle sinking and provide more accurate flux
estimates. When used from autonomous platforms at high frequency, they will also provide increased spatio-
temporal resolution for the observation of intermittent export events naturally occurring or induced by human
activities associated with marine Carbon Dioxide Removal.
**1 Introduction**
Production, transfer to depth, and remineralization of organic particles provide a major pathway for the
export of carbon from the ocean's surface to the ocean interior (Volk and Hoffert, 1985). Phytoplankton
photosynthesis and zooplankton trophic activities produce, in the sunlit ocean, particulate matter at the basis of
marine food webs. Among the different physical and biological processes determining the fate of the production,
gravitational sinking is responsible for 90% of the carbon vertical flux (Boyd et al. 2019). The mesopelagic, here
taken as the 100-1000 m layer, is also the starting depth for a myriad of processes such as particle fragmentation,
packaging and/or respiration by the mesopelagic fauna that impact (mostly reduce) the flux as particle sink (Burd
et al., 2010; Giering et al., 2014; Stemmann et al., 2004). The faster the sinking is, the more carbon is carried to a
depth where it can be stored for a long time (Boyd et al., 2019; Siegel et al., 2023).
In the common paradigm, large (>few 100's µm in size) marine particles were thought to be the main
vector of the carbon flux (Alldredge and Silver, 1988; Honjo et al., 1982; Stemmann et al., 2002). However, more
recently, other particles' characteristics than size (*e.g.*, porosity, ballasting, geometry) were assessed to be
important in setting the flux (Cael et al., 2021; Iversen and Lampitt, 2020; Stemmann and Boss, 2012; Williams
and Giering, 2022). Particles are often found in the form of aggregates composed of various lithogenic or biogenic
elements (Alldredge and Silver, 1988) and their sinking speed depends on the morphological properties of the
individual aggregates (size, density, geometry) which depends on the nature of the producers and the
aggregation/disaggregation processes (Alldredge and Gotschalk, 1988; Alldredge and Silver, 1988; Iversen and
Lampitt, 2020; Ploug et al., 2008a). The efficiency of the deep carbon sequestration depends not only on the
aggregates sinking speed but also their attenuation by various mesopelagic processes as they penetrate in the
twilight zone of the ocean. A strong vertical flux attenuation in the mesopelagic is usually observed as a result of
respiration and particle fragmentation by organisms (Burd et al., 2010; Giering et al., 2014; Stemmann et al.,
2004). On one hand, export of fast sinking particles may be less attenuated than slow-sinking ones as they spend
less time in this layer but on the other hand, they may be more prone to flux feeding by gatekeeper zooplankton
at the base of the mixed layer (Jackson and Checkley, 2011). However, this remains an open question as sinking
speed is not only dependent on size (Iversen and Lampitt, 2020) but varies with ballasting (Ploug et al., 2008b, a)
and aggregates morphology (Trudnowska et al., 2021) which are barely known.
Because the size of particles is easy to measure *in situ* with imaging systems (Gorsky et al., 2000; Picheral
et al., 2010, 2022; Stemmann and Boss, 2012) or *ex situ* in experimental design after production or collection
(Iversen et al., 2010; Ploug et al., 2010), the majority of previous studies have calculated sinking speeds, from
observations or in models, using a power law relationship between sinking and size (Burd, 2023; Forest et al.,



2013; Guidi et al., 2008; Iversen et al., 2010; Kiko et al., 2020; Kriest and Evans, 2000; Soviadan et al., 2022; Stemmann et al., 2004). However, the parameters of this size to sinking relationship vary widely as a function of plankton community composition and aggregation processes (Cael et al., 2021; Forest et al., 2013; Laurenceau-Cornec et al., 2015; Stemmann et al., 2004; Williams and Giering, 2022). In some cases, the relationship was not observed (Diercks and Asper, 1997; Iversen and Lampitt, 2020) or showed an opposite pattern (McDonnell and Buesseler, 2010) questioning the nature of the relationship between size and sinking speed or possible bias due to the experimental methods (Williams and Giering, 2022).

Sinking speeds have been generally estimated for decades experimentally, *ex situ,* on collected (or produced) material in 81% of the cases, *in situ* in specific chambers in 14% of the cases and *in situ* by divers in 5% of the cases (Cael et al., 2021; Williams and Giering, 2022). More recently, sinking speeds have been calculated by analyzing *in situ* time series of the export plume using optical or camera systems (Briggs et al., 2020; Giering et al., 2020; Stemmann et al., 2002; Trudnowska et al., 2021). There is a debate about whether *in situ* or *ex situ* provides accurate estimates of sinking speed because *in situ* ones tend to be lower (Williams and Giering, 2022). Potential bias in *ex situ* experimental design exists due to physical alteration during collection or production, or because the selected particles for experiments do not represent *in situ* particle assemblages (Williams and Giering, 2022). Conversely *in situ* methods provide an estimate of the bulk particle assemblage, possibly sorted as a function of size. The latest advancement in optical and imaging sensor technology have enabled their integration onto autonomous floats (Accardo et al., 2024; Briggs et al., 2020; Lacour et al., 2024; Picheral et al., 2022). Additionally, recent progress in unsupervised image classification now facilitates the classification of individual aggregates into categories (Accardo et al., 2024; Irisson et al., 2022; Trudnowska et al., 2021). When applied to the study of phytoplankton bloom in the Arctic Ocean, the image classification of the different types of detritus obtained using the Underwater Vision Profiler 5 deployed from a ship, showed, for the first time, a clear relationship between phytoplankton community structure and aggregates morphology with an impact on the sinking aggregate (Trudnowska et al., 2021). Networks of Biogeochemical (BGC)-Argo floats are now deployed with optical sensors to better estimate and understand processes of carbon flux and attenuation (Accardo et al., 2024; Henson et al., 2024; Lacour et al., 2024). Imaging sensors, by providing more qualitative data, will bring a substantial increase in our knowledge of mesopelagic dynamics and the interplay between particles and plankton at scales varying from local to global (Biard et al., 2016; Drago et al., 2022; Laget et al., 2024; Panaiotis et al., 2023; Stemmann et al., 2002, 2008).

In this study, we analyze the results of image analysis of a recovered Underwater Vision Profiler 6 (UVP6) camera mounted on a BGC-Argo float from May 2021 to April 2022 drifting in the sluggish flow regime of the Angola basin. During the one-year deployment, the float drifted slowly in a region with weak currents and low mesoscale activity. Seven consecutive marine snow production events were recorded at the surface and six of them lead to an export plume of sinking aggregates. Unsupervised classification of all the marine snow aggregates was performed to identify the different types of particles. The objectives of this work are i) to automatically classify *in situ* images of marine snow, ii) to describe the assemblage of marine snow particles at the surface and in the mesopelagic during the six intermittent production and export events and iii) infer marine snow morphotypes sinking velocities and vertical attenuation in the export plume. To our knowledge, this work provides the first estimates of sinking speeds and export attenuation for different types of particulate materials recorded from a BGC-Argo float.



## 2 Methods

### 2.1 BGC-Argo float deployment

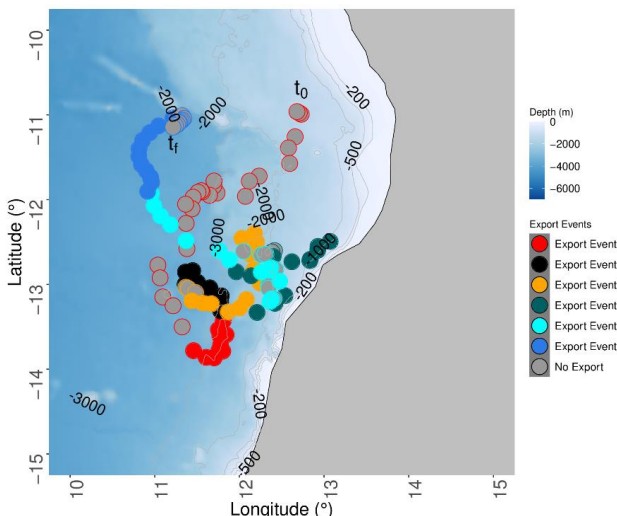

**Figure 1: Angola region of the BGC-Argo float deployment with the trajectory during the one-year drift. The filled dots correspond to the 6 export events and the colors of unfilled dots indicate the period before each export event in the same color except for the last period for which it indicates after the export. The beginning (11° S, 12°45' E noted t0) of the period before the export event1 indicates the float position at its deployment and the end of (11°10' S, 11°13' E noted tf) the period after export event6 indicates the position at its recovery.**

The BGC Argo float (WMO:6903096) was deployed on 4 May 2021 at 11° S, 12°45' E during RV Sonne cruise SO283 and retrieved on 26 April 2022 at 11°10' S, 11°13' E during RV Meteor cruise M181. Distance between surfacing every three days was on average 16 km +/- 11.73 with 4 periods (May 2021, June 2021, December 2021, February 2022) when the drift could reach 40 km in three days. In general the bathymetry was deeper than 1000 m (Fig. 1). The general pattern of the drifting was toward south, almost reaching 14° S in August 2021, then north-eastward toward the isobath 700m depth which was reached in early December 2021. Thereafter, the float drifted toward south-west along the isobath 700m until the end of December 2021. The float drifted away from the coast toward north up to the latitude 11° N at which it was recovered.

### 2.2 Environmental and satellite data

#### 2.2.1 Float data (CTD, Chla, Bbp)

The float was equipped with several sensors to characterize the properties of the water column. First, to measure hydrological parameters, the float was fitted with pressure (DRUCK_2900PSIA, SN: 11587115), temperature, and salinity sensors (SBE41CP_V7.2.5, SN: 13100). Second, to measure biogeochemical properties, it was equipped with oxygen (AANDERAA_OPTODE_4330, SN: 3489), fluorescence (proxy for Chla) and backscattering (700 nm, referred as Bbp taken as a proxy for all suspended particles ap.<10 µm) sensors (ECO_FLBB_2K, SN: 6310). Fluorescence and backscatter were converted to units of Chla (mg m$^{-3}$) and particulate organic carbon (POC, mgC m$^{-3}$) as in Accardo et al. (2024). The UVP6 (SN: 101LP) was mounted on the float. All the data were recorded during the ascent of the float, and are made freely available by the International Argo Program (https://fleetmonitoring.euro-argo.eu/float/6903095, link accessible on 06/11/2024).



**2.2.2 Satellite data (SSH, Lagrangian diagnostic)**

Several Lagrangian diagnostics were computed at each profile location using velocity data and environmental satellite products. Firstly, for each station (*i.e.*, profile location) a region considered as representative of the water column sampled by the float was defined. This region, in this study, was a circular neighborhood of radius r of 0.1° around each exact profile location. Then, this circular shape was filled with virtual particles at the surface separated by 0.01° (resulting in ~300 particles). Afterward, the next step was to compute for each virtual particle several Lagrangian diagnostics. This led to about 300 values for each sampling station which were averaged together, providing a given diagnostic around each profile location. The choice of using a circle around the float profiling location is done to smooth the errors associated with the velocity field uncertainty, as shown in previous studies (Baudena et al., 2021; Chambault et al., 2019; Ser-Giacomi et al., 2021). The velocity field used was derived from both altimetry and assimilation model delayed-time data and includes the geostrophic and the Ekman components (Copernicus Marine Environment Monitoring Service (CMEMS) product MULTIOBS GLO PHY REP 015 004-TDS). It was used to advect each virtual particle (within the representative water parcel) from the profile day until an advective time ($\tau$) ranging between 5 and 45 days backward in time. For each advective time, a diagnostic mean value was available for each profile. Diagnostics were numerous, so only those which provided significant results will be developed here. The first one used was the Finite-Time Lyapunov Exponents (FTLE, days-1). This metric is useful to identify frontal features (Baudena et al., 2021). In this study, a front is defined as a physical barrier at the surface that separates two water volumes of different hydrographic properties that likely were very far from each other in the previous days. FTLEs were calculated as in (Shadden et al., 2005) and the main parameter that was considered was the initial separation between two virtual particles. The second diagnostic used was the Lagrangian chlorophyll-a (mg m$^{-3}$), *i.e.*, the mean chlorophyll content along the backward particle trajectory. These metric estimations were computed thanks to satellite data of surface chlorophyll-a concentrations which was provided by CMEMS Copernicus website (delayed-time satellite product "OCEANCOLOUR GLO BGC L4 MY 009 104-TDS").

**2.3 Particle data taken by UVP6**

**2.3.1 A broad size classification of all particles into two size categories**

A broad size classification was applied on raw size particles spectra data (>100 μm up to few mm) provided by the UVP6 (without any plankton identification). In this case the assumption was made that zooplankton represent only a small fraction of objects sampled by the UVP6 compared to particles. Then, size range was divided into two sub-classes: MiP (Micrometric Particles) integrating the concentrations over all size classes between 0.1 and 0.5 mm and MaP (Macroscopic Particles) integrating the concentrations of size classes between 0.5 and 16 mm. This lower threshold was used because it corresponds to the definition of marine snow (Alldredge and Silver, 1988). The vertical flux of these two categories was calculated assuming an empirical relationship to convert particle size to POC and another one to obtain sinking speed from size (Kriest, 2002) (with reference 2a of Table 1 and reference 9 of Table 2 for mass and sinking speed of a particle) and previously used in the inter tropical Atlantic Ocean (Kiko et al., 2017, 2020).

**2.3.2 Unsupervised morphological classification of marine snow in four categories**

An unsupervised classification method was applied, following an approach previously used to study marine snow in the Arctic Ocean (Trudnowska et al., 2021) and in the Southern Ocean (Accardo et al., 2024).

Prior to the analysis, zooplankton and particle images were separated by supervised classification and treated independently. The first step consists in summarizing the 27 morphological features that were derived from the individual particle images. Those features (Table S1) describe their size (e.g., area, perimeter), shade intensity (e.g., mean/median gray level), shape (e.g., symmetry, elongation), and structure (e.g., homogeneity or heterogeneity, mostly based on the variability in gray level). To reach a normal distribution of each variable, extreme values (below or above the 5th and 95th percentile) were flagged as NA - which is interpreted neutral by the following methods - before applying the Yeo-Johnson transformation (Yeo and Johnson, 2000). These traits



were then summarized via dimensionality reduction using Principal Component Analysis (PCA). The PCA
function scales the features to unit variance prior to the analysis and creates a multi-dimensional "morphospace"
in which each particle was positioned based on its morphological features. As in the work of (Trudnowska et al.,
2021), size turned out to be the main morphological trait. Therefore, we repeated the same procedure by removing
all size-related features (-area, -perim., -major, -feret, -convperim, -skeleton_area, -convarea_area, -
symetrieh_area, -symetriev_area, -elongation) to hopefully better distinguish the particles by their other
morphological traits. To assess the sensitivity of this classification we also tested alternative algorithms within
this approach (see section Sensitivity of aggregates classification to the method in Supp. Mat.). In particular, we
tested UMAP (McInnes et al., 2018), a method of dimensional reduction to define the morphospace. In contrast
to PCA which creates a linear projection, UMAP is a non-linear dimensional reduction method that has been used
previously in similar contexts (Stolarek et al., 2022; Teixeira et al., 2022). To ensure comparability with the PCA,
data was scaled prior to the application and four axes were retained to define the morphospace. All other
parameters were kept with the same default values (n_neighbors=15, min_dist=0.1) since the resulting
morphospace separated the particles as expected.

The second step, the classification, was the same for all options (PCA/ UMAP, with/ without size-related
features). A k-means clustering was performed on the particles' coordinates on the first four principal components
of a morphospace**.** The number of clusters ("k") was set to four, a value which in several simulations conducted
the best trade-off between partitioning into visually clearly distinct groups of particle morphology and simplicity
in the following analysis. See section Marine snow classification in Supp. Mat. for a test of k=5 and 10 to assess
the impact of this choice on the result. Finally, the concentration (nb $m^{-3}$) of each morphotype was computed by
dividing the number of particles found in each depth bin by the volume sampled by the UVP6. To study their
spatio-temporal distributions, group concentrations were interpolated according to depth and time with a
resolution of 5 meters and one day respectively.

**2.4 Data analysis**

**2.4.1 Sinking speed and particle vertical attenuation**

We calculated the sinking speed for six export plumes that were detected by successives peaks at different
depths (Fig 2). Assuming constant sinking velocity in the upper 1000m and a Lagrangian drift in a weak vertical
shear environment during the short marine snow production events (on average <1 month), we followed the
published method developed to survey the evolution of export plumes developed for optical and imaging systems
(Briggs et al., 2020; Lacour et al., 2024), In this vertical binning approach more appropriate for large particles, a
Gaussian fit is applied to the median concentration of different aggregate size classes per 100 m depth bins. For
each size class and type of marine snow, a linear regression was done on the coordinates in depth and time of each
of the Gaussian fits' maximums. The value of the slope and the concentrations of the particles at the different
depths were stored if the Gaussian fit was successful for at least 3 depth layers. The slope is an indicator of the
sinking speed, while the particle vertical gradients were modeled using a power law model ($N(z)=N_{100}.(z/100)^{-b}$)
with a reference depth at 100 m to retrieve the $b$ exponent as an indicator of particle attenuation. This method was
initially proposed for POC flux measured in sediment traps (Martin et al., 1987) and vertical flux obtained with
profiling cameras(Guidi et al., 2008, 2015), but here it was applied to aggregate concentrations in export plumes
as in (Trudnowska et al., 2021). Due to the non-sinking behavior or because of the low signal-to-noise ratio in the
upper size range where particles are rare, this approach did not work for all size classes in all categories.

Finally, sinking speeds as a function of size and aggregate types were also calculated over the whole
deployment with lagged correlation between time series in two different depth layers (0-100 and 300-400 m). The
lag with the maximum correlation indicates the time for the different particle communities to sink from the upper
layer to the deeper layer (average distance of 300 m).

**2.4.2 Canonical Redundancy Analysis (RDA) of particle assemblage**

For each depth layer, a canonical redundancy analysis (RDA) was performed based on the abundances
of the 4 marine snow morphotypes and the above-mentioned environmental variables to explore the explanatory



power of these variables in structuring marine snow. The RDA is an extension of the multiple regression analysis
applied to multivariate data (Legendre and Legendre, 2012). It allows representing the response variables
(abundances of the 4 categories) in a "constrained" reduced space, i.e., constructed from the explanatory variables
(the environmental variables). For each RDA, the following variables were used as "supplementary variables" of
the analysis to visualize their correlation with the environmental structuring of marine snow assemblage (i.e., to
visualize their position in the RDA space). Beforehand, a Hellinger transformation was performed on the
abundances in order to reduce the impact of large concentration values. Significant axes were identified using the
Kaiser-Guttman criterion (Legendre and Legendre, 1998).
**2.4.3 Identification of intermittent production and export events in a marine snow time series**
We determined the marine snow production and export events using their time series in 5 depth layers
(Fig. 2). Major peaks were found in the different layers at a few months intervals. Except for the first surface peak
(0-100 m) in May 2021, the subsequent six surface particle peaks were followed by peaks in the mesopelagic layer
(noted 1 to 6 in Fig. 2). The beginning of each period was set at the start of marine snow accumulation in the first
layer and the end was set at the time of the subsequent minimum in the deeper layers. Throughout the manuscript,
we will refer to these six events as "plumes" or "events".

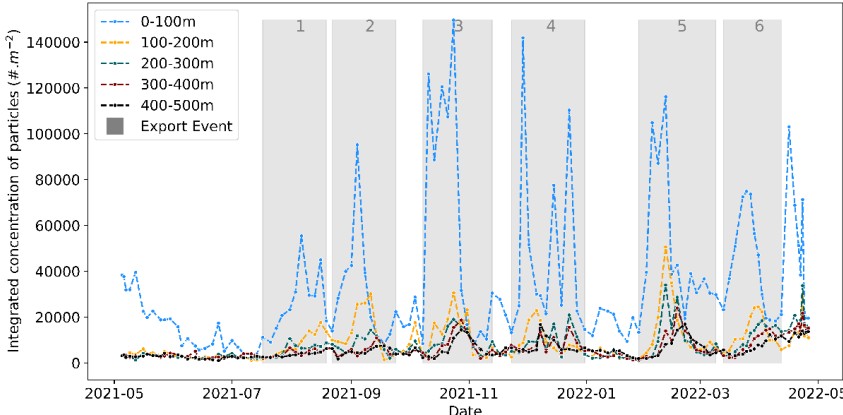

**Figure 2 : Time series of vertically integrated marine snow (MaP, all particles >600µm) concentrations (# m$^{-2}$) in 5**
**layers in the upper 1000 m depth. The 6 bloom periods shaded in gray correspond to six export events that are marked**
**by delayed peaks in the mesopelagic. The periods are defined for the events in 2021 as 07/17-08/19, 08/22-09/24, 08/10-**
**11/13, 11/23-12/31 and in 2022 as 01/28-03/09 and 03/13 to 04/12.**



# 3 RESULTS

## 3.1 Epipelagic time series of hydrological and biogeochemical properties

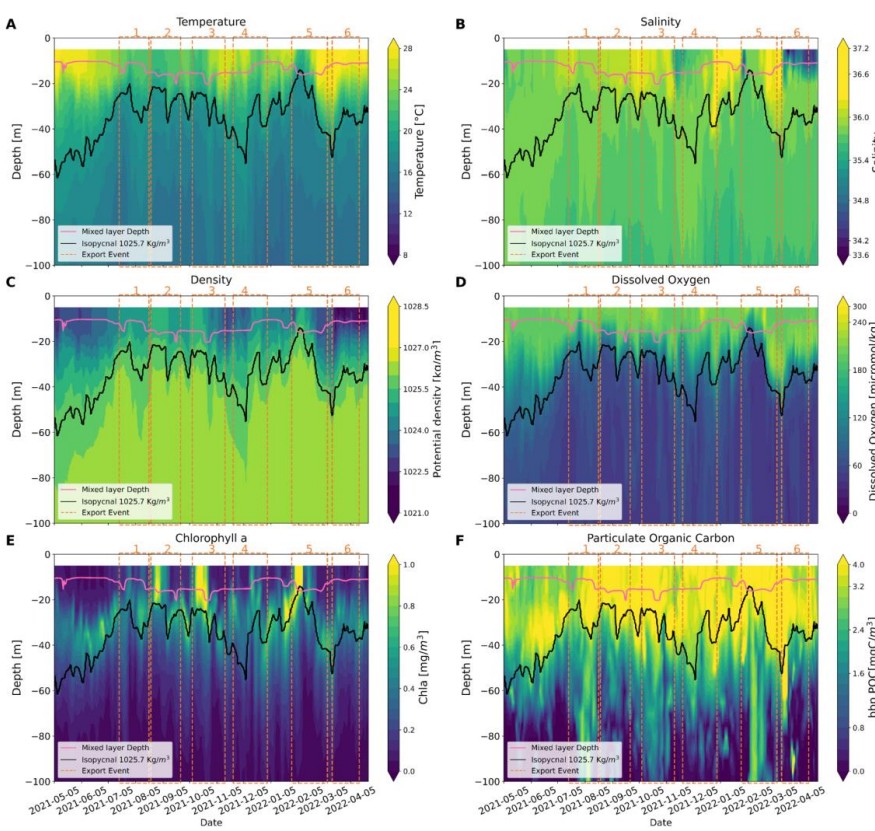

**Figure 3 : 0-100 m, 2D time series of A) temperature, B) salinity, C) density, D) dissolved oxygen,  E) chlorophyll *a* concentration and F) Bbp POC. The black line depicts the 1025.7 kg m⁻³ isopycnal. The red line represents the mixed layer depth.**

Sea surface temperature showed the lowest temperatures (21-23°C) from July to September 2021, with the highest temperatures recorded in March and April 2021 (up to 28°C) (Fig. 3). Sea surface salinity exhibited low values in November 2021 (<35) and  particularly from March to April 2022 (<34.2). Densities within the upper 30 m depth showed their lowest values from May to June, from mid-October to December 2022, and after February  2022. The isopycnal 1025.7 kg m⁻³ used as an indicator of upwelling (Körner et al., 2024), was shallow (20-30m) between July and October 2021, and as well  from December 2021 to mid-February 2022. It reached its maximum depth in May-June, December 2021 and early March 2022. Highest Chl*a* concentrations were always found in the upper 30 m with an important deep chlorophyll maximum (DCM) around 30-40 m depth. The DCM oscillated within this depth range following the rise of the 1025.7 kg m⁻³ isopycnal, displaying periods of intensification along with an upward movement towards the surface (notably in August, October 2021, and February 2022 concomitantly to the export periods 2, 3 and 5). From May to July 2021, the peak of small particles or bbp followed the DCM but thereafter to the end of the deployment (May 2022) extended from the surface down to the isopycnal 1025.7 kg m⁻³. During three periods of strong export (periods 1, 3 and 5), elevated Bbp (>3 mgC m⁻³) were observed extending down to a depth of 100 m.



**3.2 Full Depth (0-1000m) time series of Bbp, MiP and MaP**

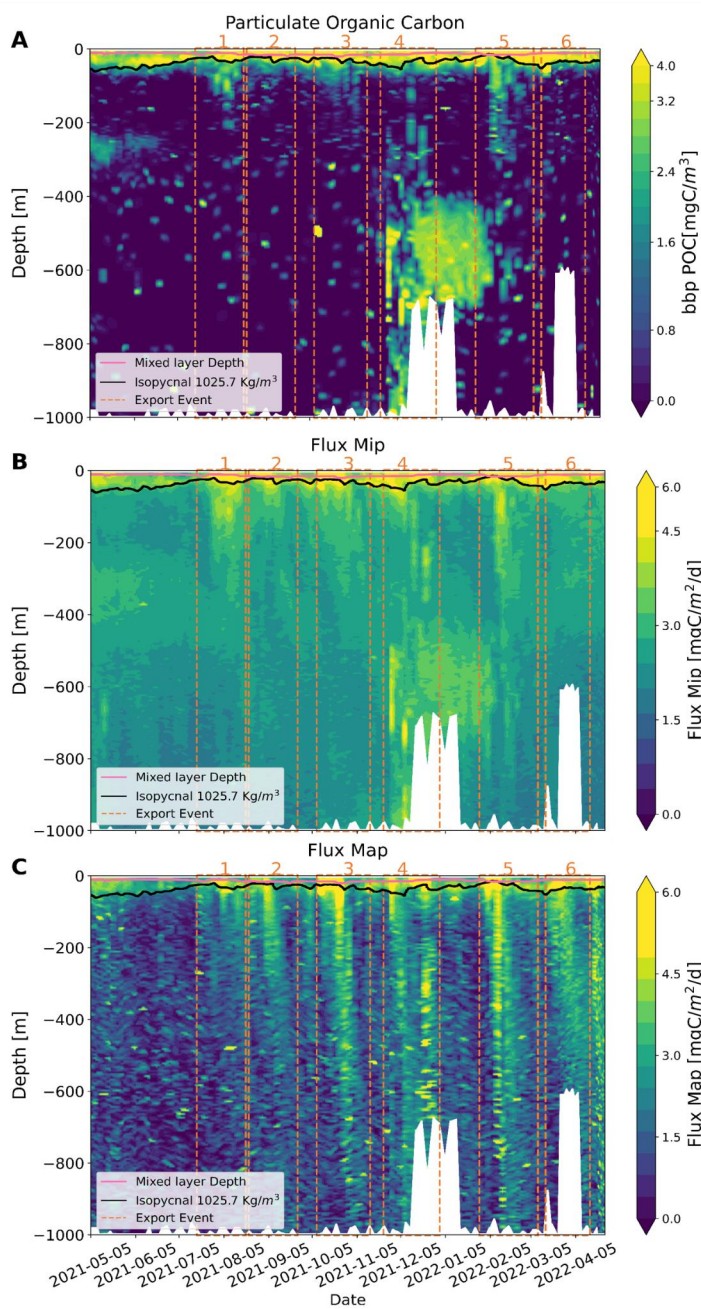

**Figure 4: Time–depth profiles determined from the BGC Argo float for A) Bbp, B) MiP flux, and C) MaP flux (in**
**logarithmic scale) as a function of time and depth with the 6 export periods being illustrated by the red vertical dashed**
**lines. The black line depicts the 1025.7 kg m⁻³ isopycnal. The maximum depth of the float that corresponds to the bottom**
**depth has been overlaid as a white mask. The red line represents the mixed layer depth.**



Small particles detected by the Bbp sensor (few µm) and the MiP (0.1<ESD<0.5mm) flux detected by
the UVP6 showed remarkably similar temporal patterns with different vertical extensions notably during the
export events. They showed all along the deployment highest values in the upper 100 m depth (Fig. 4 a-b). A
distinct midwater peak was observed from May to June 2021 between 250 and 300 m depth for Bbp and from 300
to 400 m for MiP. Very high Bbp (>20mgC m-3) and MiP (>20mg C m-2 d-1) were observed from 400 to 600 m
depth at the time when the float was in a shallow region in December 2021. Apart from this deep occurrence, the
Bbp vertical extent reached 250 m at maximum during periods 1, 3 and 5. The MiP penetrated deeper in the water
column down to 300 m depth during all export periods (albeit more in periods 1, 2, 3 and 5, August, October and
December 2021, February 2022 respectively).
In contrast, large particles as indicated by the MaP (>500 µm) flux showed a distinct spatio-temporal
pattern, with the highest values occurring in the upper 100 m depth (100-400 mgC m$^{-2}$ d$^{-1}$) differently from the
bbp and Mip which showed maximum concentration in midwater layers. Following  July 2021, the MaP flux
exhibited 6 intermittent events (>100 mgC mgC m$^{-2}$ d$^{-1}$ Fig. 4c), during which export plumes showed oblique
patterns extending from the surface down to the mesopelagic particularly in August (period 1), September (period
2), October (period 3)  and December (period 4) 2021 and February (period 5) and April (period 6) 2022. These
occurrences, notably in October 2021 and February 2022, reached depths down to 1000 m, leading to a two-fold
increase in the computed flux during the export relative to the situation before and after.
**3.3 Image classification of marine snow >600 µm into four morphotypes**

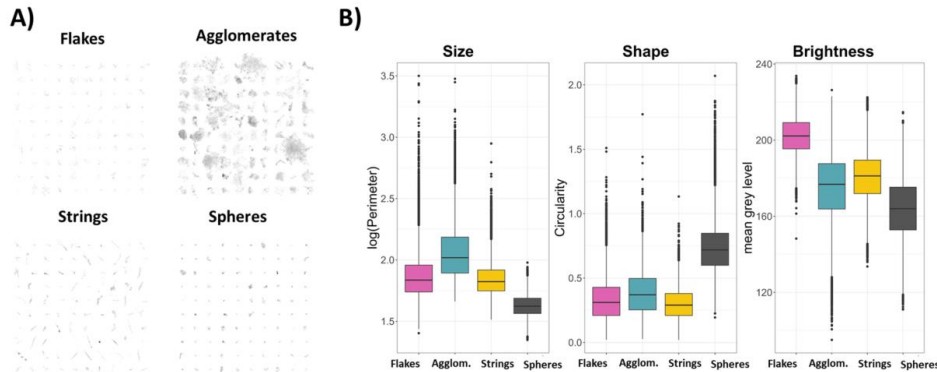

**Figure 5: A) Example vignettes of the four defined morphotypes (flakes, agglomerates, strings and spheres) and B)**
**their mean morphological traits describing main aspects of size (via perimeter (µm)), shape (via circularity,**
**dimensionless) and brightness (via mean gray level, dimensionless range 0 black to 256 white pixel).**
To depict the nature of the sinking aggregates, we classified them performing k-means clustering on the
coordinates of particles along the four retained PCA axes. Within this continuous morphospace, we identified four
morphotypes (Showed in Fig. 5A), selected as a suitable compromise between contrasted groups and contextual
knowledge to explain their nature. Cluster 1 corresponds to medium-sized and bright aggregates named **flakes**
(mean perimeter = 1.86 µm, mean circularity = 0.33, mean brightness = 202.46, Fig. 5). Cluster 2 contains large
and dark particles named **agglomerates** (mean perimeter = 2.06 µm, mean circularity = 0.38, mean brightness =
174.91). Cluster 3 contains medium-sized and elongated aggregates named **strings** (mean perimeter = 1.84 µm,
mean circularity = 0.30, mean brightness = 180.69). Cluster 4 is composed of small and circular aggregates named
**spheres** (mean perimeter = 1.63 µm, mean circularity = 0.73, mean brightness = 163.87). More results on marine
snow classification and on the sensitivity of aggregates classification to the method is provided in Supp. Mat.).



**3.4 Spatio-temporal distribution of marine snow morphotypes (>600µm)**

The different marine snow types showed concentrations of similar magnitudes varying from 0 to 20 000 particles m$^{-3}$. Surface concentrations were two to three fold higher than in the mesopelagic. From May to July 2021, all particle types showed reduced concentrations (in particular in the mesopelagic) compared to the rest of the period which showed an intermittent pattern (Fig. 6). The different particle types shared similar overall spatio-temporal dynamics mainly in the surface but also exhibited distinct features in the deep. In the surface layer and the upper mesopelagic (200-400 m), they all showed concomitant peaks during the bloom periods. In the deeper mesopelagic only agglomerates and spheres showed an increase with a time delay increasing with depth.

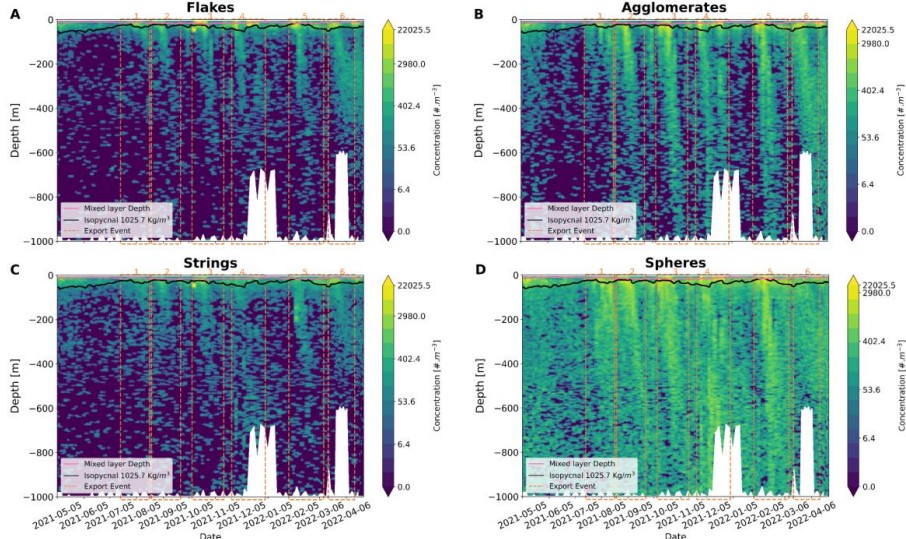

**Figure 6: Contour plot of the four marine snow types: A) Flakes, B) Agglomerate, C) Strings, D) Spheres, as a function of time and depth. The 6 export events are depicted by the red vertical dash lines. The black line depicts the 1025.7 kg m$^{-3}$ isopycnal. The bathymetry has been overlaid as a white mask.**

**3.5 Spatio-temporal dynamics of marine snow assemblage**

As illustrated in the RDA composite spaces, marine snow assemblages in the surface layer varied with seasons and also between surface and mesopelagic layer (Fig. 7). In the surface layer, contributions to axis 1, reveal the opposite dynamics between spheres and strings compared to flakes and agglomerates. Strings dominate from May to July (before export event 1), spheres from May to October (from export event 1 to export event 3), and agglomerates after February (after export event 5). The first two export events had consistent and homogeneous assemblages dominated by spheres. Subsequent export periods (3, 4, and 5) exhibited varying assemblages, mainly dominated by strings and spheres. The last period (after February) that includes export event 6 differed greatly in the assemblage with a dominance of agglomerates. When comparing periods before and during an export event in the surface layer, apart from the first event, the assemblage did not change. In the deeper layer, the assemblage composition evolution was consistent with the surface one but was more contrasted with less balanced assemblages as indicated by the higher contribution of the first axis (86.9%) relative to the surface (51.6%) (Fig. 7). In the mesopelagic, contributions to this axis reveal the opposite dynamics between spheres compared to flakes and agglomerates. Apart from export event 6, all export events were dominated by spheres. Noticeable is also a more pronounced difference between export events 2 (higher proportion of strings) and 3 (higher proportions of agglomerates and spheres) relative to the surface evolution. Dominance of flakes was not observed.




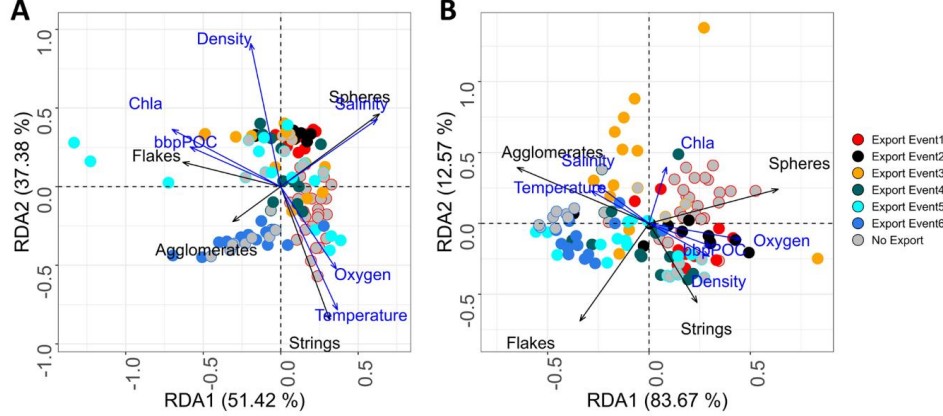

**Figure 7 : RDA of marine snow assemblages at each station (dots) A) between 0-100 m (left panel) and B) 400-500 m depth identifying the six periods of export with different colors. Stations observed prior (in gray) to the export event are encircled according to the color of the event. Black arrows show the dominant type of marine snow while the blue arrows show the correlation with environmental variables.**

### 3.6 Sinking speed and vertical attenuation of the plume

Sinking speeds, inferred from the lagged correlation between surface and mesopelagic (400-500m) particle time series, for all types of particles (MiP, MaP, morphotypes 1 to 4) were of 30 m d$^{-1}$ (Supp. Table S2). By separating the six events and with more size related classification, more details can be obtained to break apart this constant estimate. However, sinking speeds and attenuation in a plume can only be determined for a given size range if the abundance variability in the time series is consistent. This was not the case for each size class and category among the 60 possibles cases (6 plumes*10 size ranges from 100 μm to 5 mm). Sinking speeds could be estimated 6 times for flakes, 19 times for agglomerates, 5 times for strings and 12 times for spheres. When averaging altogether the results from the six export events, a unique size-based relationship is not visible for the different morphotypes (Fig. 8A), but agglomerates and spheres showed the highest sinking speeds. Also within the size range for which sinking estimates were possible for all morphotypes (ESD from 1.02-1.29 mm), spheres (46+/-24 m d$^{-1}$) and agglomerates (35+/-9 m d$^{-1}$) showed higher sinking speeds relative to flakes (16.29±4 m d$^{-1}$) and strings (18.33±6 m d$^{-1}$) (Table S3). When considering the abundance of all particles >100 μm (MiP+MaP), sinking speed estimates could be performed in 16 cases (mostly for particles larger than 500 μm). Sinking speed increased with size from a minimum of 10 m.d$^{-1}$ to a maximum of 150 m.d$^{-1}$. It is noticeable that sinking speed estimation for the largest size classes (>1.02 mm) was not possible when considering all particles but possible for agglomerates.

The strength of particle abundance attenuation in the plume increased on average with particle size and showed a remarkable difference between the different morphological types. Spheres had the lowest attenuation while flakes and strings had the strongest. For spheres, the attenuation decreased as a function of size. For size class 3, spheres (0.5+/-0.07) and agglomerates (0.88+/-0.3) had the lowest attenuation compared to flakes (1.59+/-0.02) and string (1.21+/-0.42) (Table S3). Apart from spheres, for which attenuation decreased significantly with size, no relationship between attenuation and size was found. Extending the size range by pooling all particles does not evidence an allometric relationship.



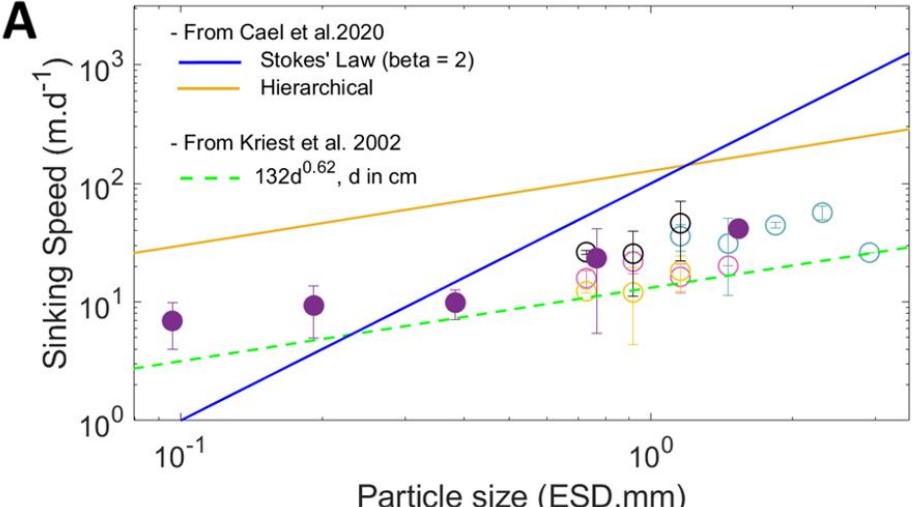

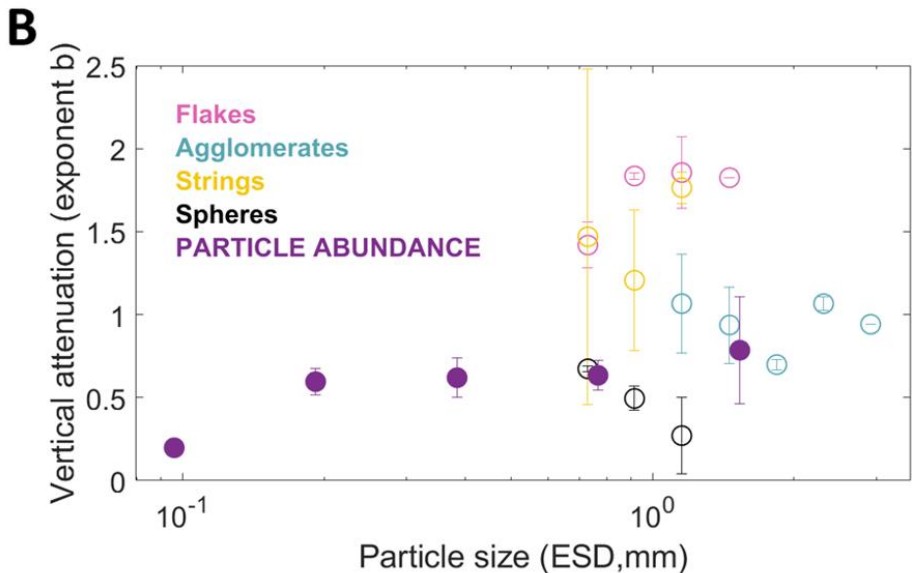

**Figure 8: Sinking speed versus size (A) and attenuation exponent *b* versus size (B) for different types of particles**
**averaged for the six different export events. The solid blue line is the Stokes' Law relationship, valid for spherical**
**smooth particles with a constant excess density with increasing size and the orange line is the hierarchical regression**
**on the data compilation in Cael et al., 2021. Dashed green line represents the model parametrizations of size-sinking**
**relationship by (Kriest, 2002).**



**4 DISCUSSION**

**4.1 Hydrological context and ocean circulation during the drift**

The observed hydrographic characteristics are in general agreement with the seasonal variability of the region. Sea surface temperature, primarily driven by surface heat fluxes, showed the lowest values from July to September and the highest values in March and April (Körner et al., 2023; Scannell and McPhaden, 2018). Sea surface salinity as measured by the float was lowest in November and from March to April. The timing of freshening is in agreement with the seasonal advection of low-salinity waters with the southward-flowing Angola Current (Awo et al., 2022; Kopte et al., 2017). While sea surface temperature shows a dominant annual cycle, upwelling and downwelling near the continental slope are characterized by both an annual and semiannual cycle. Consistent with previous study, the main upwelling season happens during the cold period from July to September while the secondary upwelling season occurs during January to February (Körner et al., 2024). During these periods, float data indicate denser water with minimum oxygen concentrations close to the surface. As there is, in general, good anti-correlation between oxygen and nitrate (Körner et al., 2024), it is possible that during upwelling periods low-oxygen and high-nitrate water enters into the euphotic zone, fueling the six observed production/export events identified in the float data. Periods 1 and 2 are within the main upwelling season and period 5 is during the secondary upwelling season. While period 3 might be during the transition from upwelling to downwelling, periods 4 and 6 were clearly during the secondary and main downwelling seasons, respectively. Note that between April and August 2021 an extreme warm event was present in the Angola basin associated with record low productivity between June and August partly covering the main upwelling season (Imbol Koungue et al., 2024).

During the deployment, the float drifted slowly with an average distance between surfacing of 16 km in 3 days. The highest drift (40 km in three days) was observed during periods 3 and 5. ADT shows weak horizontal variability suggesting that the eddy field is present but with low intensity (Fig. 4 in Supp. Mat.). FTLE values generally less than 0.1 d$^{-1}$ indicate weak eddy activity as generally reported in the region (Aguedjou et al., 2019). The negative correlation between all types of particles and FTLE indicates that the bloom events followed by particle accumulation were more intense at low horizontal mixing. The intermittent exports are associated with bloom events that are connected to coastal blooms during periods 4, 5 and to a lesser extent 6. In order to identify the marine snow export plumes (and thus calculate the sinking speeds and vertical attenuation), we made the hypothesis that the water parcel encompassing each plume did not change considerably while the float profiled them. While this Lagrangian assumption cannot be demonstrated with the available data set, different supplementary analyses suggest (Fig. S5, Table S2) that this approximation seems reasonable. The hydrological properties of the mesopelagic, as well as the oxygen concentrations, showed a weak variability during the study period. Low turbulence was observed in the region, as shown by relatively low FTLE values and by the fact that the float did not get trapped in any mesoscale eddy. In addition, previous studies showed a weak vertical shear environment in this region (Kopte et al., 2017). Delayed occurrence in large particle peaks with depth, supports this 1D hypothesis as in previous studies (Briggs et al., 2020; Lacour et al., 2024; Stemmann et al., 2002; Trudnowska et al., 2021) which is not observed in case of strong mesoscale activities (Accardo et al., 2024). A typical mesopelagic nepheloid layer (visible on Bbp and MiP), presumably extending from the seafloor was observed as the float was drifting along the 700 m depth bathymetry. This feature was also observed in the MiP but not in the larger particles (Fig. 4C and 5). Such size differentiated distribution across continental shelves has already been observed elsewhere with combined optical and imaging methods (Durrieu de Madron et al., 1990, 2017). This observation suggests that the nepheloid layers did not contain any large aggregates and therefore did not interfere with our sinking estimates for the large fractions (MaP and the 4 morphotypes) but potentially not for the MiP.

**4.2 Dynamics of the six marine snow events**

The four marine snow morphotypes were correlated to *in situ* or remotely sensed Chl*a* suggesting that most of the marine snow was of phytoplanktonic origin (Supp. Fig. S5). Additional material (*i.e.,* phytoplankton in bottles and zooplankton in net) as available in a previous study (Trudnowska et al., 2021)is needed to unambiguously attribute a morphotype to a specific phytoplankton or zooplankton community. However,



correlation with environmental variables and the shape of particles suggest possible sources of the four types of
marine snow. Typical dense fecal pellets, relative to loose marine snow, were not detected by the unsupervised
classification but they probably are mostly contained in spheres which contained the most opaque and small
particles. Given the observed lower 0-100 m integrated concentration of zooplankton organisms (observed range
$10^3$-$10^4$ ind.m$^{-2}$) relative to marine snow (observed range $10^3$- 20 $10^4$ ind.m$^{-2}$) as generally found in other studies
(Checkley et al., 2008; Forest et al., 2012; Gonzalez-Quiros and Checkley, 2006; Stemmann and Boss, 2012;
Trudnowska et al., 2021), it is possible that the contribution of fecal pellets to the total detritus was low. A lower
abundance of pellets relative to marine snow was also observed in sediment traps (Durkin et al., 2021) and episodic
export of phyto-detritus to the deep is a common feature in many sites (Turner, 2015). The three other categories
are probably mostly phytodetritus. It is likely that the two clusters of strings/filaments contained living
phytoplankton colonies as they were mostly abundant in the surface water at the time when trichodesmium
colonies were detected before the first export event (Supp. Fig. S4). While we acknowledge that the classification
in 4 morphotypes may not represent all existing morphological variability, it seems appropriate for our case study.
Four to five categories were also useful in other studies (Accardo et al., 2024; Trudnowska et al., 2021). Having
four instead of five categories increased the concentration by categories yielding more confidence in estimations
of the dynamics of marine snow assemblages, their sinking speeds, and attenuation, while still differentiating the
main morphological features of particles. In the future, global compilation of such images together with other
phytoplankton and zooplankton variables will allow to refine the number of existing morphotypes.
472         According to the float and satellite observations, the blooms seem to be triggered by different dynamics.
Blooms 1, 2, and 3 seem to be associated with typical open ocean upwelling events , as suggested by decreased
temperature and oxygen associated with higher Chl$a$ and Bbp. Conversely, the three other events (and in particular
4 and 6) were associated with less salty water. Low salinity is possibly an indicator of coastal input from particle
enriched water stemming as far off the Congo river (Brandt et al., 2023). The float was closest to the coast and
drifted along it (along the isobath 700 m depth) during period 4. During periods 5 and 6, the float drifted offshore
and surface Chl$a$ showed filaments extending from the coast (Fig. S2), corroborating the hypothesis of coastal
inputs. The bloom in period 5 was probably associated with more nutrients as suggested by the rising of the 1025.7
kg m$^{-3}$ isopycnal. The sequence of events leading to marine snow accumulation for periods 4 and 6 is less clear as
the isopycnal 1025.7 kg m$^{-3}$ was deep and Chl$a$ biomass was not so high. However, overall significant correlations
between temperature, oxygen, density, and all biogeochemical variables (Chl$a$, Bbp suspended particles, MiP,
MaP, and all zooplankton taxa; (Supp. Fig. S5) suggest that upwelled water triggered an increase in the planktonic
production.
485         Apart from the first export event, the different marine snow morphotypes did not change in their relative
contribution before and during the export events. The main changes in marine snow assemblages were associated
with depth during all events and time (the last event being dominated by agglomerates). The increase with depth
of the proportion of spheres is related to a clear reduction in filaments and flakes as previously observed in other
systems (Accardo et al., 2024; Trudnowska et al., 2021). These changes suggest that flakes and filaments are less
efficiently exported than dense particles even if they are larger in size. Although there were differences between
export events in their intensity, the general pattern of marine snow community composition, sinking and
attenuation during the first 5 events was the same (Supp. Fig. S6 and S7). Such consistency suggests their
production by phytoplankton with a constant community composition unlike the Arctic bloom which showed a
succession in primary producers types and marine snow morphotypes (Trudnowska et al., 2021). Contrasting with
the first 5 export events, the last export event showed an increased contribution of agglomerates, possibly due to
the coastal origin from the Congo river, or due to the most northern location with a different phytoplankton
community composition or because aggregation mechanisms may have been different during this event (Sup. Fig.
S3). The surface change in the community composition was mirrored in the mesopelagic layer probably resulting
from sedimentation. However, sinking speeds, nor attenuation were different from the other events.

**4.3 Sinking speed and vertical attenuation of different marine snow categories**

501         Our sinking speed estimates for 500 µm - 1 mm size range particles (10-50 m d$^{-1}$) are in the same range
as the few other estimates obtained with time series of export plume (Briggs et al., 2020; Lacour et al., 2024;
Stemmann et al., 2002; Trudnowska et al., 2021) or obtained with *in situ* devices (Diercks and Asper, 1997;
Iversen and Lampitt, 2020; Jouandet et al., 2011; Nowald et al., 2009). However, we calculate smaller sinking



speeds relative to other *ex-situ* estimates compiled in previous synthesis as illustrated by the difference between
our estimates and the hierarchical regression in the data compilation of (Cael et al., 2021) (Fig. 8). Our sinking
estimates are in good agreement with the previous model (Kriest, 2002) which was developed for miscellaneous
aggregates. Thus our results confirm that sinking speeds of natural assemblage of marine snow differ from *ex situ*
estimates, suggesting that the composition of the particles used in the experiments is biased toward fast sinking
particles (Williams and Giering, 2022).
511       Not surprisingly, dense (darker appearances implying denser and more compact structures) marine snow
particles (>500 µm) have faster sinking speed than more porous one, a property measured and modeled from
experimental works (Giering et al., 2020) but never reported for *in situ* measurement in export plume. The sinking-
to-size positive relationship was less obvious for a given aggregate morphotype because of their limited size range.
The full size range was covered only for flakes whose sinking speed remained constant with size. However, strings
and spheres had increasing sinking speeds with their size. Taken together, these two results confirm that the
sinking speed to size relationship may exist for certain types of particles but is not universal (Iversen and Lampitt,
2020; Williams and Giering, 2022) as other morphological factors (here density) are at play. Among the different
factors (phyto- and zoo- plankton community composition, inorganic ballasting) leading to denser particles
(Francois et al., 2002; Guidi et al., 2016; Trudnowska et al., 2021; Turner, 2015), our study indicates
phytoplankton aggregates to be principally responsible of the observed stronger export. Extending the size range
to 100 µm and pooling all particles together confirms the size-to-sinking relationship with however strong
variability between export plumes for the small particles and a limited size range up to the size class 1.0-1.63 mm.
For the larger size range, sinking can only be estimated after classification because non-sinking aggregates (flakes
and fibers) are dominant at the surface blurring the time series of all particles.
526       Carbon fluxes and their vertical attenuation for the different morphotypes was not calculated because the
four categories did not match the few available sizes to POC conversion factors (Alldredge, 1998; Durkin et al.,
2021). Instead, export efficiency was addressed from the vertical attenuation of the concentration of the four
different categories in the plume. Attenuation varied more with morphotypes than size. For a given size class
(1.02-1.29 mm), dense morphotypes (agglomerates and spheres) had the lowest attenuation. The exponent cannot
be directly compared to the literature because of the units but also because previous flux attenuation exponent
were calculated in a vertical frame in which depth layers are temporarily disconnected (ie, at the time of sampling
the surface layer is decoupled from the deep layers) while here we consider the number of particles in the sinking
plume as they are consumed by different mesopelagic processes (Giering et al., 2014; Stemmann et al., 2004). We
believe that following export plumes provide a more accurate estimation of the attenuation than calculating it from
vertical profiles as usually done.
537       Apart from events 4 and 5, we believe that particle spatial gradients in the mesopelagic were low and
that the large variability in the sinking speed estimates arise from other methodological factors that are important
to discuss to improve further studies. First, the quality of the sinking estimates depend on the abundance of
particles *in situ*. Compared to conditions with massive seasonal blooms over a long period of several months
(Lacour et al., 2024), the production and export events in the Angola basin were less than 1.5 months with on
average 10 times lower concentrations. Both factors yielded a patchy spatial distribution that was smoothed by
using a depth bin of 100 m instead of varying depth bin (20 to 200 m) (Lacour et al., 2024). This high variability
was amplified as large particles were subdivided in four morphotypes. Second, the deployment in the Angola
basin took place close to coastal upwelling systems. Offshore propagating filaments may have had effects on
particle vertical distribution mainly on small particles (here Bbp and MiP). Third, the clustering method is not
sufficiently selective to obtain homogeneous groups among the aggregates as the overlaps can be seen on the PCA
space (Supp. Fig. S1). This non-perfect classification has the effect of smoothing the derived estimates of sinking
speed and attenuation.



**5 Conclusion**

We describe seven bloom events leading to surface accumulation of marine snow based on data obtained from a UVP6 camera mounted on a BGC-Argo float and recovered after one year of deployment. Six of them led to an export event with different types of aggregates and different penetration depths. For the first time, two core parameters for carbon sequestration, sinking speeds, and vertical attenuation, were calculated *in situ* for different sizes and morphotypes. Not all detected marine snow aggregates are sinking despite them being larger than 1 mm. Within a given size range, we show that sinking speeds of porous marine snow are smaller than that of dense marine snow indicating the strong impact of density on sinking speed. However, we show that size is still an important property to determine the sinking speed when considering a larger size range or a specific type of marine snow. Compared to earlier studies which could not distinguish aggregates morphology, the proposed classification allowed us to calculate sinking speeds of millimetric marine snow even in the case when a large fraction of them were not sinking. Compared to published synthesis on marine particles sinking speeds, our *in situ* estimates are consistent with empirical allometric models parameterized for marine aggregates and lower than most *ex situ* estimates posing the question of the impact of the methodology. This study demonstrates the high potential of using cameras on autonomous floats to assess intermittent export following episodic bloom events. To better understand particle dynamics and better assess carbon flux, future works should improve the following key methodological issues, 1) a global library of marine snow images to develop classification algorithm adapted for regional and global applications, 2) size to POC conversion factor for the different types of aggregates, 3) couple BGC-Argo floats with ship surveys to provide more comprehensive contextual data than only those derived from the float and satellite data as in this study, 4) increase acquisition frequency to detect rare larger particles, 5) implement embedded recognition in camera because most of the floats are not recovered.

**Author Contribution Statement**

YDS, LS, RK, designed the study
YDS, MB, JH, AB, RL, LS, AA, PB, LD worked on the different components of the data analysis
AB calculated the Lagrangian metrics
YDS and LS drafted the manuscript
All authors reviewed the manuscript.

**Competing interests**

The contact author has declared that none of the authors has any competing interests

**Acknowledgment**

The authors acknowledge the support of the crew of RV Sonne during cruise SO283 and RV Meteor during cruise M181 for the deployment and recovery of the BGC-Argo float. We are thankful to Marc Picheral and Camille Catalano and the Plateforme d'Imagerie Quantitative de Villefranche (PIQv) to make the UVP6 data available. RK acknowledges funding from the Heisenberg Programme of the German Science Foundation #KI 1387/5-1. RK and AB acknowledge support via a "Make Our Planet Great Again" grant of the French National Research Agency within the "Programme d'Investissements d'Avenir"; reference "ANR-19-MPGA-0012". AB, RK, LS, SSp, LD, PB acknowledge support via EU H2020 grant (agreement 817578 TRIATLAS project). AB acknowledges support via "SEASONS" project of the French National Center of Spatial Studies (CNES). LD acknowledges support via Sorbonne Université through the Ecole doctorale 129. YDS acknowledges support via "Make Our Planet Great Again" visiting fellowship program for early career researchers and granted by the French Ministry for Europe and Foreign Affairs, in collaboration with the French Ministry for Higher Education and Research, and implemented by Campus France.



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
