# Peer review of "Marine snow morphology drives sinking and attenuation in the ocean interior"

_EGUsphere, 2024_

## Author Response (AR1)

Reviewed comments of the manuscript «**Marine snow morphology drives sinking and attenuation in the ocean interior**» MS No.: egusphere-2024-3302

**Responses to the Reviewers' comments**

Answers to reviewers' comments are reported point by point. The questions and comments of the reviewers are in black, **the answers in black (bold),** *the text added to the manuscript in italic*

**RV1: Citation: https://doi.org/10.5194/egusphere-2024-3302-RC1**

**First we would like to warmly thank Reviewer 1 for her/his relevant, constructive comments and positive feedback which helped to improve the manuscript.**

The study makes a significant contribution to oceanographic research by addressing key gaps in our understanding of marine snow dynamics, particularly its morphology and sinking speeds. The use of in situ measurements from a BGC-Argo float equipped with advanced optical sensors represents an innovative approach, providing real-time, high-frequency data that enhance both spatial and temporal resolution.

The scientific methodology is robust, combining in situ measurements, unsupervised classification, and a thorough analysis of environmental variables. The integration of UVP6 imaging with biogeochemical sensors enables a detailed investigation of particle dynamics, surpassing traditional size-based relationships.

The manuscript is well-organized, with clear sections that guide the reader through the methodology, results, and implications. Figures and tables are informative;

-Overall Evaluation

The article is a strong contribution to the field, combining methodological innovation with significant findings that enhance our understanding of marine snow dynamics. Addressing the following recommendations would further strengthen the impact and clarity of the manuscript:

The questions and comments of the reviewer 1 are in black, **the answers in black (bold),** *the text added to the manuscript in italic:*

-Argo Data Usage

Please clarify where the Argo float data originate from. Did you use a DOI monthly snapshot from ARGO? (http://www.argodatamgt.org/Access-to-data/Argo-DOI-Digital-Object-Identifier).

**Response: The plankton and particle data obtained from the UVP sensors were downloaded from Ecopart (https://ecopart.obs-vlfr.fr/#) and Ecotaxa (https://ecotaxa.obs-vlfr.fr/). The images were retrieved directly from the UVP6's memory card and exported to Ecotaxa for quality control and for image classification in planktonic taxa and morphological categories.**

**BGC-Argo float data were collected through the International Argo Program and can be found at https://argo.ucsd.edu. We considered the NetCDF files from http://doi.org/10.17882/42182#117069 (float with WMO 6903096, link accessible on 12/03/2025)**

Specify which Argo data were utilized and confirm whether ADJUSTED data were used when available.

**Response: Thank you for your question. We used ADJUSTED data whenever available to ensure the highest data quality and consistency. When ADJUSTED data were not available, we relied on REAL-TIME or DELAYED MODE data as appropriate.**

Clearly state which quality control (QC) procedures were applied to the data. This is essential to ensure the reliability and reproducibility of the analysis. If any modifications or deviations from standard Argo QC protocols were made, they should be explicitly described.

**Response: Thank you for your question. We did not apply additional QC procedures beyond those already implemented in the standard Argo QC protocols. We calculated derived variables as in Accardo et al., 2024. We will make this explicit in the manuscript by modifying the text.** *(See in revised manuscript the end of the paragraph '2.2.1 Float data')*

**Otherwise, analyses were conducted using the quality-controlled data as described in the Methods section.**

-Argo Program Acknowledgment

The provided link (https://fleetmonitoring.euro-argo.eu/float/6903095) is incorrect and is not the proper ARGO link for the data. Please ensure correct referencing of the Argo Program.

**Response: We apologize for this typographical error. It will be corrected in the revised manuscript. Here is the new link: https://argo.ucsd.edu and we corrected it in the manuscript.**

-Include the following acknowledgment for Argo data and its DOI:

**Thank you for these inputs. They will be added as suggested in the revised manuscript:**

*"Argo data were collected and made freely available by the International Argo Program and the national programs that contribute to it. (https://argo.ucsd.edu, https://www.ocean-ops.org). The Argo Program is part of the Global Ocean Observing System.*

*Reference: Argo (2000). Argo float data and metadata from Global Data Assembly Centre (Argo GDAC). SEANOE. http://doi.org/10.17882/42182"*

-Technical Corrections

Lines 157–161: The serial numbers of the sensors are not a key point; please consider removing them.

**Response: We agree with reviewer 1, information about the serial numbers of the sensors are not important. The inclusion of sensor serial numbers was intended to enhance traceability and provide a clear reference for data verification. However, we understand that they may not be essential. Following reviewer 1's recommendation, we will remove the serial numbers of the sensors in the submitted manuscript.**

Line 295 and Figures 3, 4: Use Bbp, not bbp.

**- "bbp" has been corrected to "Bbp" in the revised manuscript.**

When referencing "Bbp POC," ensure consistent terminology throughout the manuscript.

**- The terminology "Bbp POC" has been standardized throughout the manuscript for consistency.**

Figures 3, 6, and 5A are not very clear, possibly because they are too small.

**- These figures have been adjusted for better clarity in the revised manuscript (see the new figures attached).**

Reviewed comments of the manuscript «**Marine snow morphology drives sinking and attenuation in the ocean interior**» MS No.: egusphere-2024-3302

**Responses to the Reviewers' comments**

Answers to reviewers' comments are reported point by point. The questions and comments of the reviewers are in black, **the answers in black (bold)**, *the text added to the manuscript in italic*.

**Responses to the comments of the anonymous Reviewer 2**

**First we would like to warmly thank the anonymous Reviewer 2 for her/his relevant and constructive comments which helped to improve the manuscript.**

**RV2: Citation: https://doi.org/10.5194/egusphere-2024-3302-RC2**

General comments

The study by Soviadan et al. describes an innovative application of the UVP particle imager to characterize and quantify particle fluxes in a coastally influenced ecosystem. This study represents a significant advancement in how the biological pump can be monitored because they are incorporating biological diversity of particles into their flux and sinking estimates. This approach has been developed by earlier studies, but the application during a sustained, temporally resolved dataset generated by a BGC-ARGO float is particularly exciting and new. The authors detected 6 export events, as inferred by the downward movement of a plume of sinking particles. From these observations, they could resolve differences in sinking speed and flux attenuation among particles with different morphologies and densities. These data help explain discrepancies in size vs. sinking speed relationships observed by others. These observations reiterate the importance of biological diversity among sinking particles for quantifying carbon export by the biological carbon pump. The ability to sustain these observations of time is also particularly exciting, because the largest export event often occur as episodic events that are missed by ship-based observations.

Overall, I think this is a strong manuscript.

**Thank you very much for your thoughtful and positive feedback on our manuscript.**

Specific comments:

In figures 2 and 6, you are using particle concentration to track the timing of the events or the downward movement of the plume with depth. Since this just adds up the particles across size bins, isn't this essentially the same as only reporting the abundance of particles in the smallest size bin, because they are exponentially more abundant than the larger sizes? The smallest sized particles will dominate any numerical abundance. Would a different pattern emerge if you assessed these events using either 2-D surface areas of the ROIs, or an estimated biovolume? Using a measure like biovolume would account for the influence of large, rare particles. The current analysis is most strongly influenced by the abundance of the smallest size bin, without accounting for changes in the slope of the particle size distribution. I think the authors need to explicitly acknowledge this bias in the data and justify their choice, or else change the way these data are analyzed and displayed.

**Response: Thank you for your comments and your proposal that we have included in the revised version.**

**Figure 2 reports the concentrations of all objects >600µm while figure 6 reports the concentrations of the different morphotypes. The reviewer is right that they depict mostly the particles in the small size range which are more numerous. However, we show spatio-temporal patterns of the different small (<600µm) and large (>600µm) particles in Figure 4. The morphotypes vary considerably in their size (see Figure 5) so that size is implicitly taken into account in Figure 6. In addition, to calculate the sinking speed of the different morphotypes, we calculate the time series for the different size for all particles and also for the four morphotypes (5 layers*5 types of particles*10 size class) yielding the generation of 100 possibles figures instead of only 2).**

**We attached the same figure as Figure 2, for biovolume and the 6 events are also observed with more variability due to the occurrence of rare large aggregates. Two additional events with a lower surface intensity can be detected. Surface peaks of 455.33 mm$^3$.m$^{-2}$ between events 2 (ap 1976.67 mm$^3$.m$^{-2}$) and 3 (ap. 13849.80 mm$^3$.m$^{-2}$) and surface peak of 1011.51 mm$^3$.m$^{-2}$ between events 4 (ap 9554.23 mm$^3$.m$^{-2}$) and 5 (ap. 7902.77 mm$^3$.m$^{-2}$). These surface peaks in volume are followed by deeper delayed peaks but with fewer points. We tried to include them in the peak detection algorithm used to assess the sinking speed and they did not pass the quality control of having more than 3 points to calculate the linear regression. So we did not include them in the sinking speed analysis and we added a small paragraph to explain that the 6 export plumes were the main ones and that two minor ones could be detected.**

**We added the additional Figures based on particle biovolumes (see the two last Figures 2S and 6S in the attached files: New Figures).**

**We added to the text in the section 2.4.3 the following:**

*Throughout the manuscript, we will refer to these six events as "plumes" or "events". Apart from these 6 main peaks, we observed between period 2 and 3 and between period 4 and 5, two small peaks in surface concentrations that translated into weak peaks in the deeper layer for the first one. Converted to biovolume (Figure S2), these two small events were more visible suggesting that they were composed of mainly large rare particles. They were not included in further analysis because they did not met the quality control to estimate sinking speeds (see section 3.4.1) due to the more noisy time series of the rare large objects.*

In the legend of figure 4, the legend indicates that fluxes are displayed in log scale, but the scale bar appears to be in linear units. This needs some clarification.

**Response: We apologize for this typographical error. The legend of Figure 4 has been clarified to accurately reflect the scale used. See the new figures attached for the corrected version.**

It is difficult to see the particles in figure 5. Can the contrast be increased? I can't distinguish the difference between these morphotypes.

**Response: we increased the contrast in the manuscript and we specified the changes we made (see the new figures attached)**

The methods have very little information on the float and how its mission was programmed. For example, in figure 4 it says the white areas are the float depth and the bottom. In Figure 5 it says the white areas are the bottom depth. The methods indicate that the depths were typically deeper than 1000m. Was the float programmed to float immediately above bottom? How far above bottom?

**Response: We acknowledge the lack of clarity and have reworded the text. The maximum depth of the float corresponds to the bottom depth, which is indicated by a white mask.**

**The text was changed to:**

*In general the float was programmed to reach a maximum depth at 1000 m and when possible every 4 profiles to reach 2000 m (Fig. 1). When the bottom depth was shallower than 1000m the float maximum depth was set to be 20 m above seabed.*

On line 391 you refer to "size class 3". What is this size class? In the text on this line, it is not clear exactly how attenuation is related to spherical size. Does attenuation decrease with increasing size or decreasing with decreasing size? I can see the relationship in the figure, but the text is ambiguous.

**Response: Thank you for your comment and your questions**

**"Size class 3" refers to particles with a spherical equivalent diameter ranging from 1.02 mm to 1.29 mm as reported in the tableS3 in the supp. material. This will now be clarified in the manuscript, with a reference to the appropriate table.**

**Regarding attenuation: the strength of particle abundance attenuation in the plume, an increase as a function of size is not observed in the limited size range. There was more an effect of the morphology (dense particles are less attenuated).**

**We modified the text to make it clear:**

*Abundance attenuation for the four morphotypes in the plume did not increase on average with particle size and showed that attenuation varied more with morphotypes than size. For a given size class (1.02-1.29 mm), dense morphotypes (agglomerates and spheres) had the lowest attenuation suggesting more efficient transfer to the deep ocean.*

[Figure]

Reviewed figures (and additional figures only for reviewer2) of the manuscript «**Marine snow morphology drives sinking and attenuation in the ocean interior**» MS No.: egusphere-2024-3302

**Figure 2: Time series of vertically integrated marine snow (MaP, all particles >600μm) concentrations (# m$^{-2}$) in 5 layers in the upper 1000 m depth. The 6 bloom periods shaded in gray correspond to six export events that are marked by delayed peaks in the mesopelagic. The periods are defined for the events in 2021 as 07/17-08/19, 08/22-09/24, 08/10-11/13, 11/23-12/31 and in 2022 as 01/28-03/09 and 03/13 to 04/12.**

[Figure]

**Figure 3 : 0-100 m, 2D time series of A) temperature, B) salinity, C) density, D) dissolved oxygen, E) chlorophyll *a* concentration and F) Bbp POC. The black line depicts the 1025.7 kg m⁻³ isopycnal. The red line represents the mixed layer depth.**

**3.2 Full Depth (0-1000m) time series of Bbp POC, MiP and MaP**

[Figure]

Figure 4: Time–depth profiles determined from the BGC Argo float for A) Bbp POC, B) MiP flux, and C) MaP flux (in logarithmic scale) as a function of time and depth with the 6 export periods being illustrated by the red vertical dashed lines. The black line depicts the 1025.7 kg m$^{-3}$ isopycnal. The maximum depth of the float corresponds to the bottom depth, marked by a white mask. The red line represents the mixed layer depth.

[Figure]

**Figure 5: A) Example vignettes of the four defined morphotypes (flakes, agglomerates, strings and spheres), we applied microsoft windows photo software changes to the vignettes with contrast (-40%), brightness (-8%), sharpness (100%) and B) their mean morphological traits describing main aspects of size (via perimeter (µm)), shape (via circularity, dimensionless) and brightness (via mean gray level, dimensionless range 0 black to 256 white pixel).**

[Figure]

**Figure 6: Contour plot of the four marine snow types: A) Flakes, B) Agglomerate, C) Strings, D) Spheres, as a function of time and depth. The 6 export events are depicted by the red vertical dash lines. The black line depicts the 1025.7 kg m⁻³ isopycnal. The maximum depth of the float corresponds to the bottom depth, marked by a white mask.**

**Additional Figures based on particle biovolumes for Review only**

[Figure]

**Figure S2: Time series of vertically integrated marine snow (MaP, all particles >600µm) biovolume (mm⁻³m⁻²) in 5 layers in the upper 1000 m depth. The 6 bloom periods shaded in gray correspond to six export events that are marked by delayed peaks in the mesopelagic. The periods are defined for the events in 2021 as 07/17-08/19, 08/22-09/24, 08/10-11/13, 11/23-12/31 and in 2022 as 01/28-03/09 and 03/13 to 04/12.**

[Figure]

**Figure S6: Contour plot of the four marine snow types based on particle biovolumes: A) Flakes, B) Agglomerate, C) Strings, D) Spheres, as a function of time and depth. The 6 export events are depicted by the red vertical dash lines. The black line depicts the 1025.7 kg m⁻³ isopycnal. The maximum depth of the float corresponds to the bottom depth, marked by a white mask.**